# Enhanced Performances of Quantum Dot Light-Emitting Diodes with an Organic–Inorganic Hybrid Hole Injection Layer

Ling Chen [1,*], Donghuai Jiang [1], Wenjing Du [1], Jifang Shang [1], Dongdong Li [2,*] and Shaohui Liu [1,*]

1   Henan Key Laboratory of Electronic Ceramic Materials and Application, College of Materials Engineering, Henan University of Engineering, Zhengzhou 451191, China
2   Huaibei Yeolight Technology Co., Ltd., Huaibei 235000, China
*   Correspondence: cling555@163.com (L.C.); lidongdong@yeolight.com (D.L.); qqliushaohui@163.com (S.L.)

**Abstract:** PEDOT:PSS (polyethylene dioxythiophene:polystyrenesulfonate) is a commonly used hole injection layer (HIL) in optoelectronic devices due to its high conductive properties and work function. However, the acidic and hygroscopic nature of PEDOT:PSS can be problematic for device stability over time. To address this issue, in this study we demonstrated the potential of an organic–inorganic hybrid HIL by incorporating solution-processed WOx nanoparticles (WOx NPs) into the PEDOT:PSS mixture. This hybrid solution was found to have a superior hole transport ability and low Ohmic contact resistance contributing to higher brightness (~62,000 cd m$^{-2}$) and current efficiency (13.1 cd A$^{-1}$) in the manufactured quantum-dot-based light-emitting diodes (QLEDs). In addition, the resulting devices achieved a relative operational lifetime of 7071 h, or approximately twice that of traditional QLEDs with PEDOT:PSS HILs. The proposed method is an uncomplicated, reliable, and low-cost way to achieve long operational lifetimes without sacrificing efficiency in optoelectronic devices.

**Keywords:** WOx-nanoparticle-doped PEDOT:PSS; hole injection layer; quantum dot; light-emitting diode

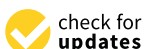



## 1. Introduction

Quantum dot light-emitting diodes (QLEDs) are becoming increasingly popular as a potential choice for future light-emitting applications. This is mainly due to their unique features such as tunable colors, high color purity, low-cost solution processability, and inherent photophysical stability [1–5]. The first successful demonstration of QLEDs was conducted by A.P. Alivisatos [6], and since then the external quantum efficiency (EQE) has improved significantly, reaching values above 30%. This has been made possible by the advancement of process technology and the accumulation of knowledge on materials and device architectures [7–9]. The most advanced QLED structures currently available incorporate a multilayer design that sandwiches the QD emitting layer between an inorganic electron transport layer and an organic hole injection/transport bilayer (HIL/HTL). The HIL usually consists of poly(3,4-ethylenedioxythiophene):polystyrene sulfonate (PEDOT:PSS) due to its high work function, low-temperature solution processability, high transparency, and conductivity [1,4,7–9]. However, despite achieving high efficiency, the moisture- and acid-sensitive nature of the PEDOT:PSS HIL often leads to device degradation, thereby limiting performance and operational stability [10,11].

To address the issues with PEDOT:PSS, solution-processed transition metal oxides such as NiO, MoO$_3$, WOx, CuO, and V$_2$O$_5$ have been explored as alternative HILs in QLEDs due to their high stability in heat, hydrogen, and oxygen [12–18]. An example of these advancements can be seen in Yang et al.'s early studies [12], which investigated the use of WOx nanoparticles (NPs) as a hole injection layer (HIL) instead of PEDOT:PSS in constructing green quantum dot light-emitting diodes (QLEDs). This resulted in a peak EQE of 3.22% and nearly a two-fold increase in operational lifetime compared to QLEDs with traditional PEDOT:PSS-HIL. Ding et al. [13] also demonstrated the efficacy of using a solution-processed copper oxide (CuO) film as a HIL for QLEDs, achieving an

EQE of 5.37% and significantly improved device stability. Zhang et al. [15] investigated NiO films as HILs for QLED devices, which showed a six-fold increase in operational lifetime compared to traditional PEDOT:PSS HIL-QLEDs. Zeng et al. [16] devised a novel solution using a scraper coating technique to create a hole injection layer (HIL) composed of solution-treated molybdenum oxide (MoOx). Through this method, they were able to customize the morphology, composition, and surface electronic structure of the MoOx HIL by incorporating isopropanol in the precursor during the coating process. The resulting MoOx HIL was implemented in a QLED, demonstrating a remarkable lifetime of 3236 h for the red device, double that of conventional PEDOT:PSS HIL. While these metal oxides have demonstrated improved device stability, there are still challenges to overcome such as energy-level mismatch and poor crystallinity [19–21], which can limit the luminescent performance of devices.

To address these challenges, bilayer PEDOT:PSS [19,20,22,23] and transition-metal-oxide-doped PEDOT:PSS [24–26] have been successfully adopted in the field of various optoelectronics. In fact, organic–inorganic hybrid HILs have shown promise for improving QLED stability and performance. Kim et al. [19] have reported on an ultra-thin tungsten oxide/PEDOT:PSS mixed hole extraction layer that capitalizes on the benefits of PEDOT:PSS and addresses the device stability concerns of PEDOT:PPS. This technological development has resulted in a marked improvement in the stability of P3HT:PC$_{60}$BM organic solar cells, with a modest enhancement in power conversion efficiency also observed. In our previous work [20], we demonstrated that a WO$_3$/PEDOT:PSS bilayer HIL can improve lifetime without sacrificing efficiency. Lee et al. [25] utilized a combination of PEDOT:PSS and MoO$_3$ as a modification layer for the anode. By incorporating MoO$_3$, the PEDOT:PSS surface was not only rendered more easily wettable but also demonstrated improved conductivity. This resulted in more effective hole injection, ultimately achieving a maximum current efficiency of 1.39 cd A$^{-1}$ with their inverted QLED, outperforming the standard device by 27%. Shin et al. also reported success using PEDOT:PSS:V$_2$O$_5$ films as the HIL prepared by using a sol–gel method, resulting in all solution-processed QLEDs with maximum luminance and current efficiency values of 36,198 cd m$^{-2}$ and 13.9 cd A$^{-1}$, respectively [26]. Therefore, the development of new organic–inorganic hybrid materials is a promising method for advancing the field of QLEDs.

In this study, we demonstrated that a high-quality organic–inorganic hybrid HIL possessing superior hole transport ability and low Ohmic contact resistance can be easily formulated by mixing a precise amount of solution-processed WOx nanoparticles (WOx NPs) into PEDOT:PSS. By incorporating WOx NPs, we were able to achieve full solution-processed QLEDs with significantly improved luminance and current efficiency that also exhibited a longer device lifetime. Our findings highlighted that using the organic–inorganic hybrid HIL in QLEDs is a cost-effective and reliable method with uncomplicated experimental procedures to achieve longer operational lifetimes without compromising efficiency.

## 2. Materials and Methods

We first conducted the synthesis of the green alloy structure CdSe@ZnS, ZnO, and WOx NPs. As per our previous works [27], the green CdSe@ZnS QDs were synthesized. To prepare WOx NPs, the hydration method reported previously [28] was followed. In this method, hydrochloric acid solution was slowly added to an aqueous ammonium metatungstate hydrate solution with stirring under a constant rate until the pH of the mixed solution was between 1 and 1.25 at room temperature. Subsequently, the solution was heated and stirred for 1 h. Prior to use, the WOx solution was filtered using a 0.45 mm membrane filter. The colloidal ZnO NPs were synthesized by following a reported method [1]. The as-synthesized ZnO NPs were washed twice with heptane, and the particles were finally dispersed in ethanol for our experiment (approx. 30 mg mL$^{-1}$).

In the fabrication of the QLED device, firstly, ITO-coated glass substrates with a sheet resistance of approximately 15 Ω sq$^{-1}$ were sequentially cleaned in ultrasonic baths containing detergent, deionized water, acetone, and isopropanol for 15 min each. The

substrates were then exposed to an ultraviolet ozone ambient for 15 min. After this cleaning procedure, PEDOT:PSS solutions or WOx:PEDOT:PSS solutions with different blend ratios were spin coated onto the substrates at 5000 rpm and baked at 150 °C for 15 min. The substrates were then transferred to a $N_2$-filled glovebox for the deposition of TFB, QDs, and ZnO NP layers. TFB was dissolved in chlorobenzene at a concentration of 8 mg mL$^{-1}$, spin coated at 3000 rpm, and annealed at 150 °C for 30 min. The green QDs were dispersed in chloroform at a concentration of 15 mg mL$^{-1}$ and then spin coated at 3000 rpm for 60 s. Next, the ZnO NPs were deposited at a speed of 1500 rpm as electron transport layers (ETLs). The multilayer samples were then loaded into a high-vacuum deposition chamber where the Al cathode (100 nm) was thermally deposited at a pressure of $\leq 1 \times 10^{-6}$ mbar. Finally, the QLED devices were encapsulated using ultraviolet-curable resin and cover glasses.

For characterization, transmission electron microscope (TEM) images of QDs and WOx NPs were recorded by using a JEOL JEM-2100 electron microscope. The absorption spectra of both the QDs and WOx solutions were measured using a UV–vis spectrometer (Lambda 950, PerkinElmer, Waltham, MA, USA). The PL spectra were collected using a spectrofluorometer (JY HORIBA FluoroLog-3). A scanning electron microscope (SEM) (Nova Nano SEM 450) and atomic force microscopy (AFM) (Dimension Icon) were used to obtain the surface topography images. The current density–luminance–voltage (J–V–L) characteristics and electroluminescence spectra of the QLEDs were measured under ambient conditions using a Keithley 2400 sourcemeter and a PhotoResearch PR-735 spectrometer.

## 3. Results and Discussion

### 3.1. Characterization of WOx NPs

The morphology and size of the synthesized WOx NPs were studied using TEM. As shown in Figure 1a, the WOx NPs were spherical particles with an average diameter of approximately 20 nm. However, the WOx NPs tended to aggregate due to interactions between them, highlighting the need for the introduction of suitable ligands to improve their dispersion in future. The absorption spectrum in Figure 1b indicates that the WOx solution retained a high transmission across the entire visible spectrum. This was crucial for our thin-film QLED designs, which will be discussed later in this article. Moreover, the band gap energy (Eg) of the WOx NPs could be determined by the relation $\alpha h\nu = (h\nu - Eg)^m$, where m = 1/2 for direct bandgap semiconductors [29]. The linear section of the curve (Figure 1b, inset) could be extrapolated to the zero absorption coefficient, allowing the estimation of Eg as 3.8 eV. The XRD patterns of the as-prepared and annealed WOx films are shown in Figure 1c. The as-prepared WOx film was amorphous, but the film became crystalline at 300 °C with various stoichiometric states forming.

The composition of the WOx films was analyzed using X-ray photoelectron spectroscopy (XPS) spectra, which are illustrated in Figure 2. The O 1s peak was found at 530.1 eV. The XPS spectrum of the W 4f core level, as shown in Figure 2b, revealed peaks at 33.2 eV (W 4f$_{7/2}$) and 35.4 eV (W 4f$_{5/2}$) that could be attributed to the high oxidation state of W in the WOx. Additionally, a broad peak of 5p3/2 was observed at approximately 39.3 eV. As per previous literature [30,31], these peak positions and shapes were designated to represent the WOx compound in $W^{6+}$.

### 3.2. QLED Performance

The green alloy structure CdSe@ZnS QDs were selected as the emitting layer because they possessed a spherical shape with an average diameter of approximately 10 nm (as illustrated in Figure 3a). The UV–vis and PL characteristics of the CdSe@ZnS QDs were examined, with the PL peak observed at 519 nm without any surface defect-related emissions (as demonstrated in Figure 3b). Photographs of the excited QDs in methylbenzene are included in the insets. The difference in wavelengths between the absorption and PL spectra band maxima was primarily caused by the Stokes shift [32].

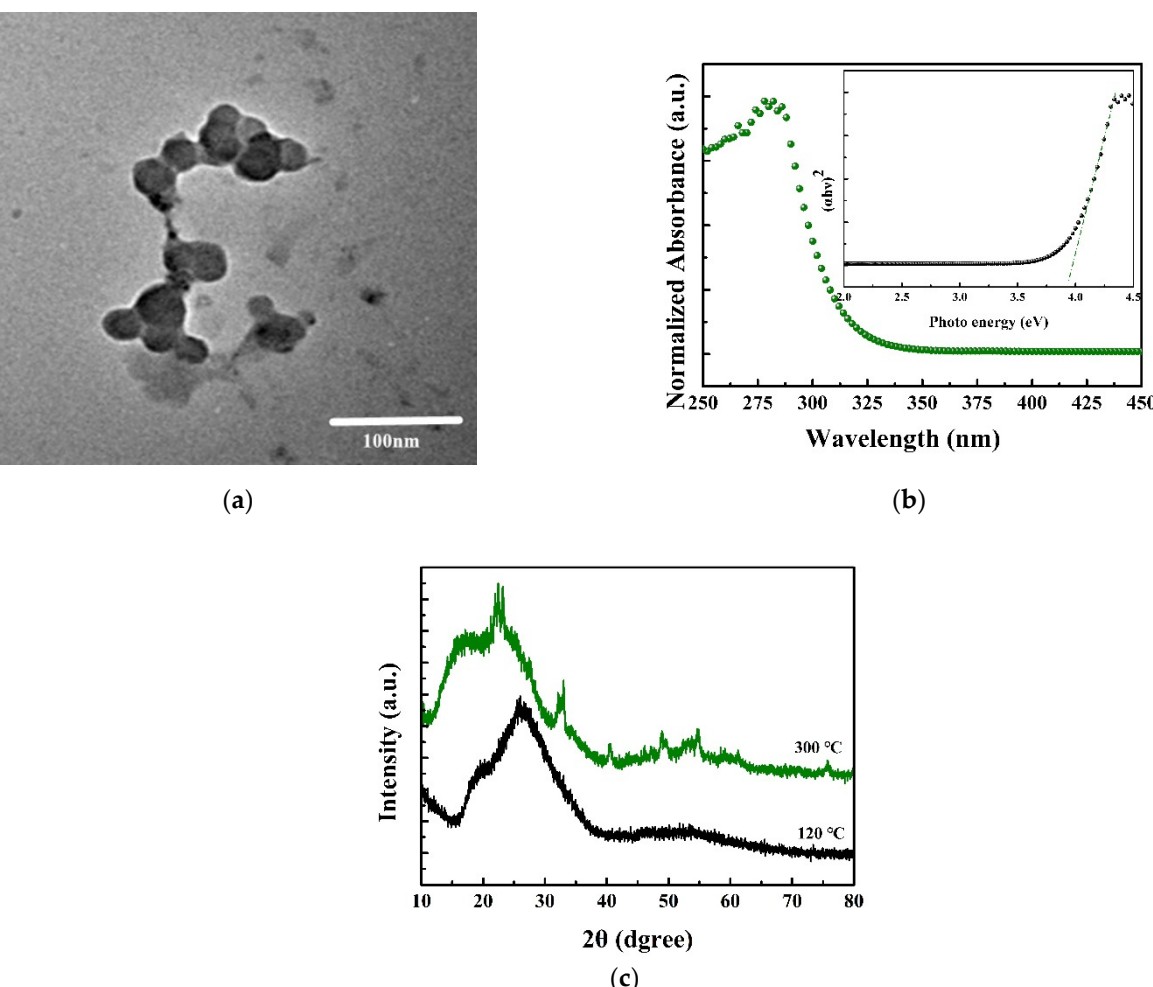

**Figure 1.** (**a**) TEM micrographs of the as-prepared WOx NPs. (**b**) Ultraviolet–visible absorption spectrum of WOx film. The inset shows a plot of the absorption coefficient vs. the photon energy for the determination of the WOx optical band gap. (**c**) XRD patterns of WOx films (annealed at 120 °C and 300 °C, respectively).

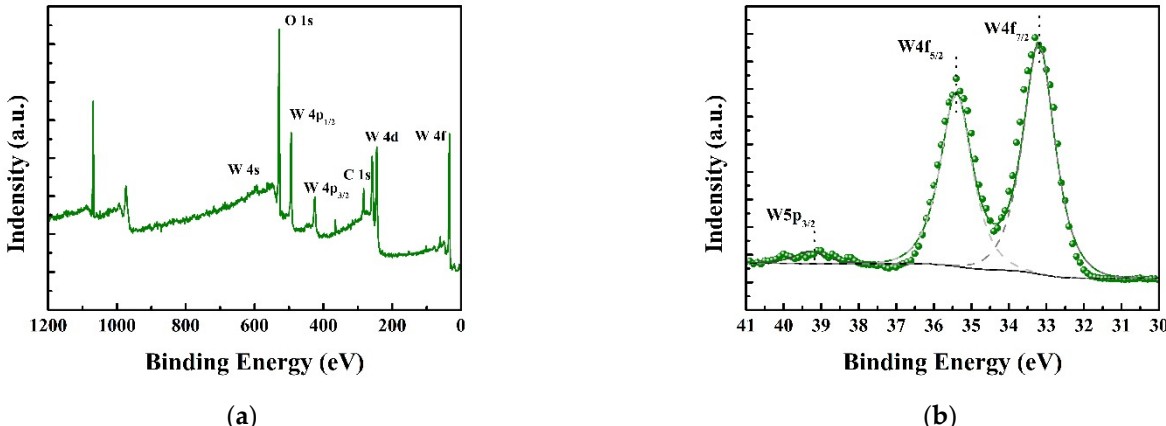

**Figure 2.** (**a**) Wide-scan XPS spectrum of the WOx films; (**b**) W 4f XPS spectrum of the WOx and its peak fitting.

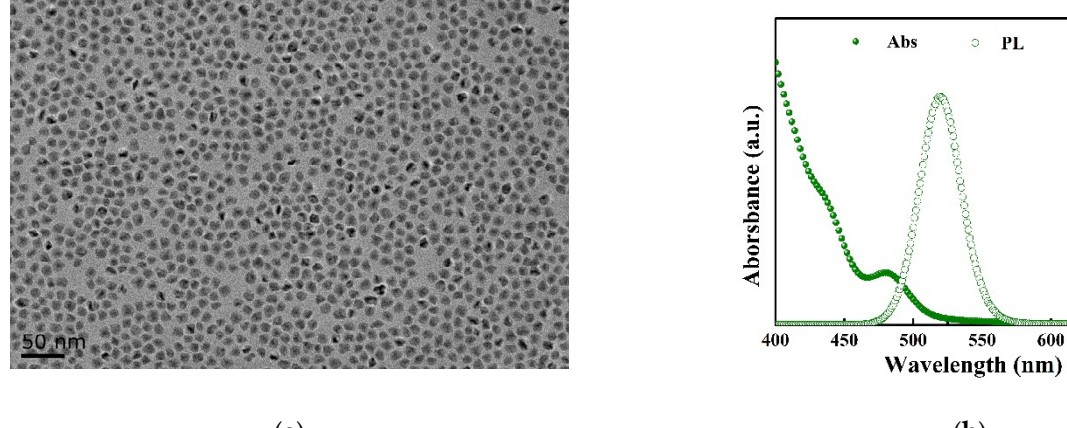

**(a)**                                                                                          **(b)**

**Figure 3.** (**a**) TEM image of CdSe@ZnS QDs; (**b**) PL and absorption spectra of the QDs in methylben-zene (the inset shows the image of the QDs dispersed in methylbenzene).

In this study, a thin film was formed by spin coating and annealing a mixture of PE-DOT:PSS and tungsten oxide in a certain proportion, which acted as a HIL for QLEDs (as illustrated in Figure 4a). The control QLED device had a multilayer architecture (Figure 4b) consisting of an ITO anode, a PEDOT:PSS or WOx:PEDOT:PSS HIL, a TFB hole HTL, a green QD EML, a ZnO nanoparticle ETL, and an Al cathode. Based on the energy level diagram (Figure 4c), the injection of electrons into the QD emissive layer in QLEDs was slightly easier compared to the injection of holes due to a smaller energy barrier between the ZnO NPs and QDs. The addition of the WOx layer further strengthened the hole injection in the green CdSe@ZnS QLEDs, resulting in a shift of the major recombination center toward the QDs/ZnO interface and a significant enhancement of the recombination rate [33]. The WOx layer effectively acted as an electric dipole layer with electrons diffusing from the PEDOT:PSS and TFB to the deep-lying conduction band of WOx (~6.5 eV) and leaving holes in the PEDOT:PSS and TFB.

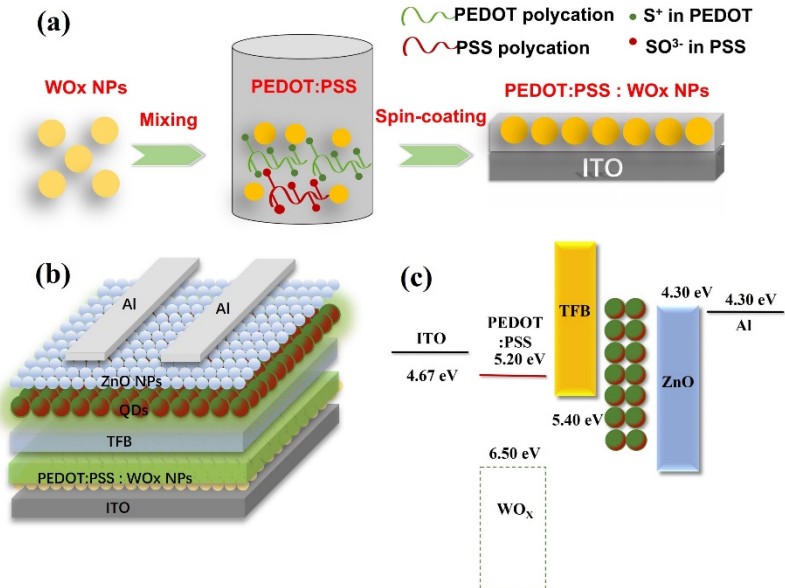

**Figure 4.** (**a**) Schematic diagram of the preparation and formation process of WOx:PEDOT:PSS films; (**b**) structure of QLED and the functional layer materials used in this work; (**c**) schematic illustration of the energy level diagram.

The J−L−V characteristics and current efficiency−power efficiency with luminance (CE−PE−L) of the QLEDs are presented in Figure 5a,b, with the volume ratio of PEDOT:PSS

and WOx ranging from 5:1 to 15:1 to determine the optimal ratio. Table 1 summarizes the performance parameters of QLEDs using mixtures of PEDOT:PSS and WOx as HTLs. In Figure 5a, it is evident that the doping of WOx NPs resulted in an increase in luminance. In devices with a 15:1 ratio of PEDOT:PSS:WOx HIL and 10:1 ratio of 10:1 PEDOT:PSS:WOx HIL, the luminance values were significantly enhanced, with maximum values improving from 46,000 cd m$^{-2}$ in the control device to 52,430 cd m$^{-2}$ and 62,000 cd m$^{-2}$, respectively. These improvements corresponded to 14.0% and 34.7% enhancements, respectively. However, when the volume ratio of PEDOT:PSS to WOx was increased to 5:1, the efficiency of the device decreased instead of improving, indicating that an excessive amount of WOx added could actually reduce the luminous efficiency of the device. Figure 5a shows that both the devices had a low turn-on voltage of ~2.7 V, indicating efficient carrier injection into the QD layer. The enhanced hole injection could improve the charge balance and ultimately enhance the device's efficiency, as demonstrated by the CE and PE curves in Figure 5b. For instance, the standard device achieved the maximum CE of 9.8 cd A$^{-1}$ at a voltage of 3.8 V. In contrast, the device incorporating 10:1 PEDOT:PSS:WOx HIL achieved the same value of CE with just 3.5 V and exhibited a maximum CE increase of approximately 33.7%, reaching 13.1 cd A$^{-1}$. It is worth noting that these devices demonstrated impressive reproducibility, as evidenced by Figure 5c. In fact, the average current efficiency of 16 devices was measured at 11.09 cd A$^{-1}$, exceeding that of the reference device. Additionally, while standard QLEDs exhibited the maximum PE of 6.5 lm W$^{-1}$, devices with 10:1 PEDOT:PSS:WOx HIL achieved a maximum PE of 8.9 lm W$^{-1}$, indicating a 36.9% increase compared to pure PEDOT:PSS. This improved performance was attributed to the introduction of WOx, which increased the carrier mobility and the probability of electron–hole radiative efficiency within the QD layers. In the same vein, it is worth mentioning that as the volume ratio of PEDOT:PSS to WOx was increased to 5:1, the current efficiency of the device dropped to 5.1 cd A$^{-1}$. This highlighted that the device efficiency was quite sensitive to the quantity of wo added. If the dosage exceeded a certain limit, it could negatively impact the quality of the device's film.

**Table 1.** Performance parameters of the standard and optimal QLEDs.

| Device | $V_T$ (V) | $L_{max}$ (cd m$^{-2}$) | $\eta_P$ (lm W$^{-1}$) | $\eta_A$ (cd A$^{-1}$) | FWHM (nm) |
|---|---|---|---|---|---|
| Control | 2.7 | 46,000 | 6.5 | 9.8 | 38 |
| 10:1 | 2.6 | 62,000 | 8.9 | 13.1 | 38 |

Figure 5d illustrates the EL spectra of the standard and optimal QLEDs at 3.5 V. The spectra reveal a Gaussian-shaped peak with a narrow FWHM of 38 nm, similar to the PL spectrum of the QDs at a slightly longer wavelength (around 4–5 nm). This was attributed to the Förster resonant energy transfer (FRET) from smaller to larger dots within the QDs [34]. Additionally, the inset in Figure 5d shows an EL photographic image of a QLED fabricated using our CdSe@ZnS QDs. The bright green emission from this QLED was due to the excellent current injection of the CdSe@ZnS QDs, which emitted brilliantly.

Compared to the PEDOT:PSS HIL, the most significant advantage of using a PEDOT:PSS:WOx film was the significant improvement in the operating lifetime of the device. The stability of a PEDOT:PSS:WOx-based QLED and a PEDOT:PSS-based device were compared, and tests were conducted in air with a simple UV-curing epoxy resin encapsulation, as shown in Figure 6. Both devices underwent testing with a constant driving current density of approximately 200 mA cm$^{-2}$. Correspondingly, the initial brightness of the PEDOT:PSS-HIL device was 3800 cd m$^{-2}$, while the PEDOT:PSS:WOx-HIL device had an initial brightness of 5000 cd m$^{-2}$. The PEDOT:PSS:WOx-based QLED demonstrated a half-life time of approximately 20 h. This corresponded to a long device lifetime of approximately 7071 h at an initial luminance of 100 cd m$^{-2}$ ($L_0^n T_{50}$ = constant, n = 1.5, which represented the acceleration factor) [10]. This indicated an almost two-fold increase in the operating lifetime compared to the PEDOT:PSS-based QLED, which displayed an operating lifetime of only 3162 h. The enhanced stability of the device could be primarily attributed

to the superior thermal stability of inorganic hole transport materials in comparison to their organic counterparts. Furthermore, the device based on PEDOT:PSS:WOx exhibited a higher efficiency and carrier recombination rate when subjected to a luminance of 5000 cd m$^{-2}$, which ultimately translated to reduced damage from accumulated carriers.

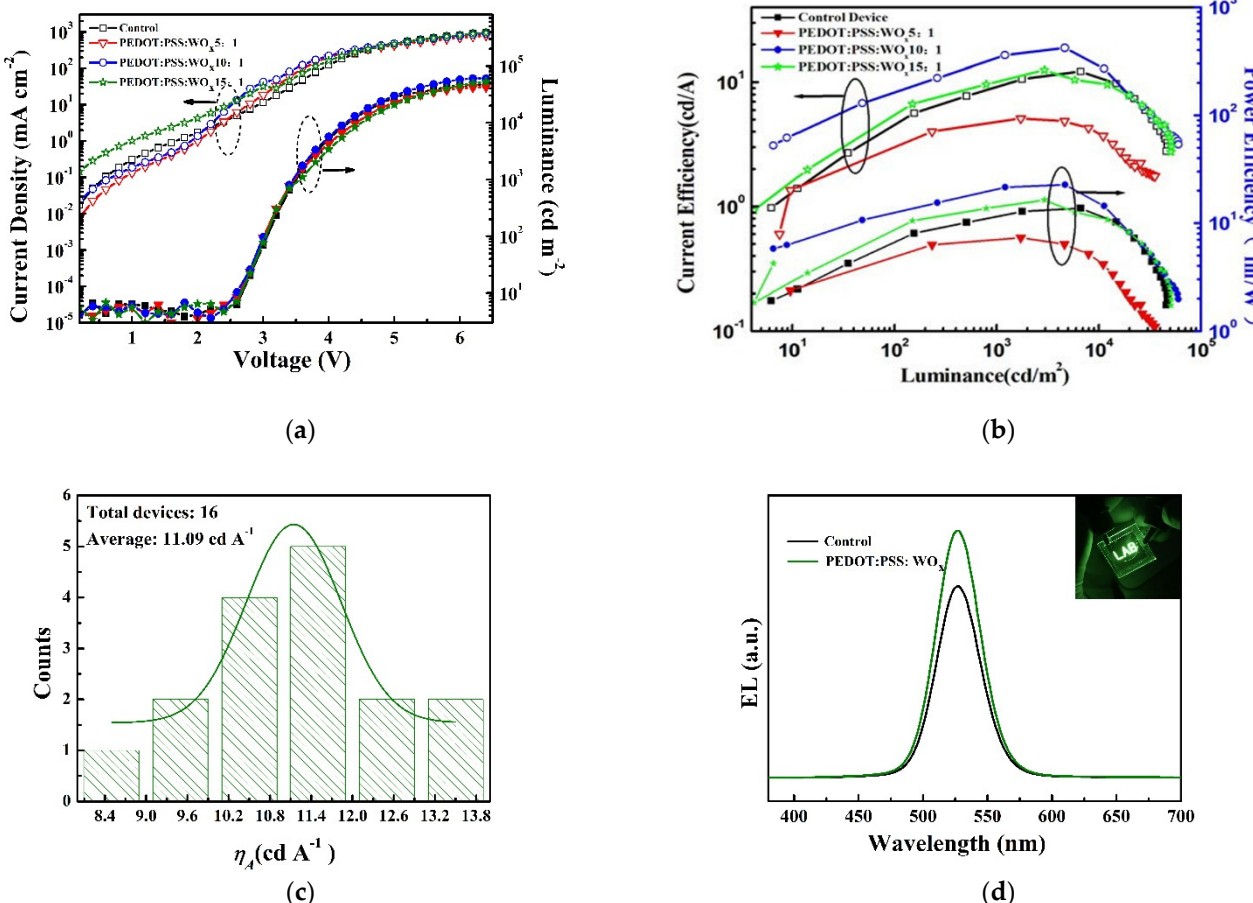

**Figure 5.** (**a**) Current density−luminance−voltage (J−L−V); (**b**) current efficiency−power efficiency with luminance characteristics for the different volume ratios of PEDOT:PSS:WOx−based QLED; (**c**) quantity statistics of the maximum current efficiency of the device with optimal HIL; (**d**) EL spectra of the standard and optimal device (the inset shows a photographic image of the green QLED).

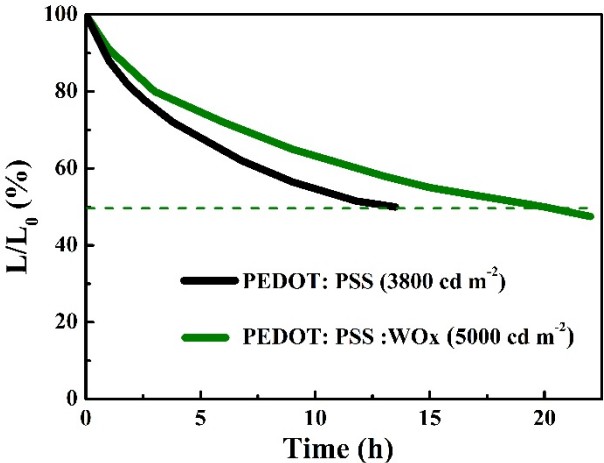

**Figure 6.** Operating lifetime characteristics of the PEDOT:PSS:WOx−based and PEDOT:PSS-based QLEDs.

### 3.3. Effect of the Introduction of WOx on the Optics, Electricity, and Morphology of the PEDOT:PSS Film

It is commonly accepted that the morphology of the HIL (or HTL) plays a crucial role in the performance of the all-solution-processed QLED, and a smooth surface can effectively reduce the occurrence of micro electrical shorts [20]. Therefore, surface morphology measurements were conducted using both scanning force microscopy (SEM) and atomic force microscopy (AFM). As shown in Figure 7a–c, the SEM images illustrate the surface morphology of the ITO glass, ITO/PEDOT:PSS, and ITO/PEDOT:PSS:WOx (10:1), respectively. It is apparent that the ITO/PEDOT:PSS had a well-formed film that exhibited some clefts, as shown in Figure 7b. However, the use of a PEDOT:PSS:WOx layer as the anode buffer layer led to a better surface morphology compared to ITO/PEDOT:PSS, exhibiting smaller clefts. This indicated that the deposition of the PEDOT:PSS:WOx layer altered the surface morphology of the anode. Figure 7d–f display 3DAFM images at the 2 μm × 2 μm scan size of the samples. Through surface analysis of Figure 7d using software, it could be concluded that the root mean square (RMS) roughness of the pure ITO surface was 2.2 nm. After the PEDOT:PSS:WOx layer was added, the RMS roughness was reduced to 1.02 nm and the surface became much flatter, which was comparable to the PEDOT:PSS-modified ITO substrate (with an RMS roughness of 1.25 nm, as shown in Figure 7e). This modification of the surface roughness by WOx resulted in low Ohmic contact resistance, serving as a strong foundation for constructing QLEDs with highly smooth and clear HTL and QDs layers.

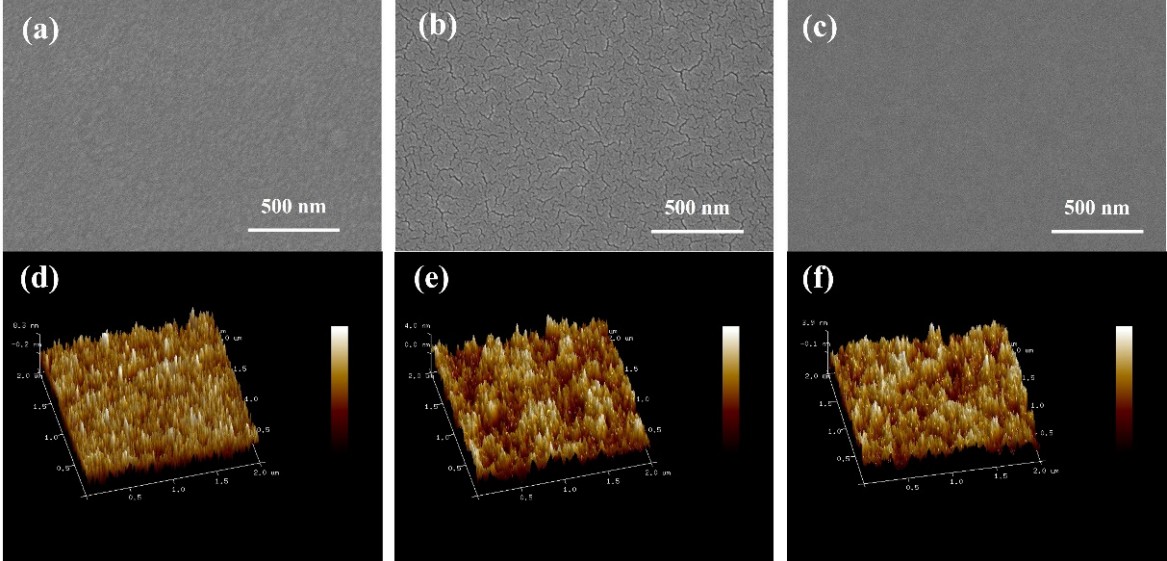

**Figure 7.** SEM images of (**a**) the ITO glass and the (**b**) ITO/PEDOT:PSS and (**c**) ITO/PEDOT:PSS:WOx (10:1) films; AFM images of (**d**) the ITO glass and the (**e**) ITO/PEDOT:PSS and (**f**) ITO/PEDOT: PSS:WOx (10:1) films. The surface roughness RMS values for each condition were 2.20 nm, 1.25 nm, and 1.02 nm, respectively).

On the other hand, in order to produce high-quality QLEDs, a high level of transparency in the HIL is required. To assess this, we also measured the transmittance of the ITO/PEDOT:PSS and ITO/PEDOT:PSS:PSS:WOx substrates. As depicted in Figure 8, the results indicated that all of the substrates exhibited similar levels of transmittance across the visible spectrum, with consistently high levels (~90%). In the UV region, however, the ITO/PEDOT:PSS:PSS:WOx substrate showed a slight reduction in light transmittance that was attributed to the bandgap transitions of the WOx film.

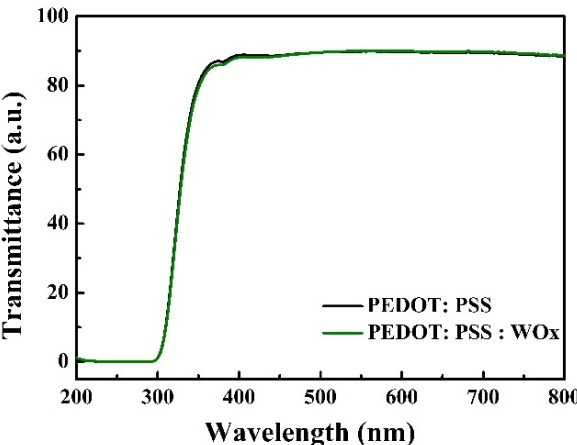

**Figure 8.** Transmittance of the PEDOT:PSS/glass and PEDOT:PSS:WOx/glass.

To investigate the hole injection and transport capability of QLEDs with both PEODT:PSS and PEDOT:PSS:WOx structures, Au/HTL/HIL/ITO hole-only devices were analyzed. In this device structure, the large injection barrier at TFB/Au should obstruct electron injection from the Au electrode to the TFB. As a result, measuring the J–V characteristic curves could effectively characterize the hole injection and transport capability. Figure 9 displays the J–V characteristic curves of the TFB/PEDOT:PSS and TFB/PEDOT:PSS:WOx hole-only devices with Au and ITO electrodes at both ends. The J–V curves of the hole-only devices exhibited comparable electrical characteristics to those of QLEDs. Furthermore, the hole-only devices containing TFB/PEDOT:PSS:WOx demonstrated an increased current density at high voltages. This finding suggested that incorporating PEDOT:PSS:WOx enhanced the conductivity of holes in QLEDs at elevated voltages, improving their electrical and electro-luminescent performance.

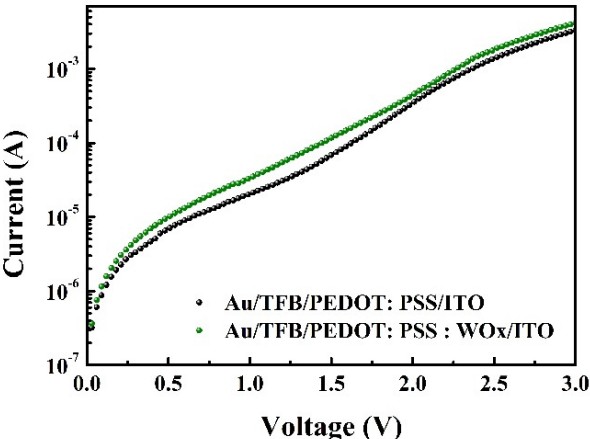

**Figure 9.** Current density−voltage characteristics of the hole-only devices with pure PEDOT:PSS and PEDOT:PSS:WOx (10:1) HILs.

## 4. Conclusions

The findings of this study suggest that using WOx-doped PEDOT:PSS as a hole injection layer can significantly enhance the stability and performance of QLEDs. The QLEDs demonstrated an impressive improvement of approximately 36.9%, which could be attributed to the enhanced hole injection/transmission performance and better Ohmic response of the PEDOT:PSS:WOx mixture with a volume ratio of 10:1 compared to the pure PEDOT:PSS. The use of stable WOx instead of problematic PEDOT:PSS resulted in a significant improvement in the stability of the devices. This approach offers a simple, cost-effective, and versatile method to enhance the efficiency and stability of QLEDs.

**Author Contributions:** L.C. and D.L.: writing—original draft and writing—review and editing; J.S. and D.J.: methodology; W.D. and S.L.: formal analysis. All authors have read and agreed to the published version of the manuscript.

**Funding:** This research was funded by the National Natural Science Foundation of China (grant number 52202168); the Program for Science & Technology Innovation Talents in Universities of Henan Province (grant number 21HASTIT014); the Excellent Youth Fund of Henan Natural Science Foundation (grant number 212300410031); the Natural Science Foundation of Henan Province, China (grant number 202300410098); the Scientific Research Foundation of the Higher Education Institutions of Henan Province, China (grant numbers 21B140002, 23A140019, and 22B430009); and the Henan University of Engineering Foundation, China (grant number DKJ2019012).

**Data Availability Statement:** Not applicable.

**Conflicts of Interest:** The authors declare no conflict of interest.

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
