# Peer review of "Enhanced Performances of Quantum Dot Light-Emitting Diodes with an Organic–Inorganic Hybrid Hole Injection Layer"

_crystals, doi:10.3390/cryst13060966_

Round 1

Reviewer 1 Report

The authors report on the effects of WOx nanoparticles inclusion in PEDOT-PSS on the performances of quantum dots LEDs.

The paper is well organized ancd clearly written. However it should be improved by addressing the following comments:

1) The WOx effects are basically described by comparing the LED without WOx with the best device (with content 10:1). The other two LEDs, with 5:1 and 15:1 content are not discussed. Looking ant the luminance curve the devices without WOx, with 5:1 and 15:1 contents look basically identical. The authors should comment on why no improvements on the luminance are observed for these two WOx contents. In addition, looking at the power and current efficiency it is evident that the 5:1 LED behaves worse than both the reference and the 10:1 sample. This result is rather strange and needs a comment. Moreover, as just one device (the reported 10:1) seems improved with respect to the standard LED, the reproducibility of the results should be discussed by reporting average figures of merit for all the devices, and not just one number.

2) Concerning the lifetime the authors ascribe the higher stability of the 10:1 sample to improved thermal stability. However the measurements are perofmed at the same luminance, thus the 10:1 sample is measured at lower current. The higher stability could thus simply come by lower thermal degradation due to the lower heating of the device, without the need of different thermal stability.

3) Section 3.3 should be renamed, as it does not discuss the effect of WOx content on the LEDs performances (discussed in the previous section), but the effect of WOx content on the hole transporting layer morphology. Moreover the SEM and AFM measurements are inverted in the discussion. For the AFM measurements it would be useful to use the same z scale, in order to simply evidence the different roughness.  Concerning the AFM image of pedot-pss without WOx, the image doesn't "shows that the root mean square (RMS) roughness of the ITO surface is 2.2 nm", the image shows the morphology from which one can determine the roughness.

Reviewer 2 Report

The authors present an article about Enhanced Performances of Quantum Dot Light-Emitting Diodes using a Organic–Inorganic Hybrid Hole Injection Layer.

The article is written in a very clear manner, presenting both the fabrication steps and the results in a comprehensive way. The effect of the material used is evident and appears to be very promising for future device applications. The figures are also clear. Therefore, I have no specific criticisms for the authors. I would only suggest expanding the introduction and the related bibliography regarding the existing literature on QD-based LEDs, as it currently seems too concise.
